# Role of Metformin in Morbidity and Mortality Associated with Urinary Tract Infections in Patients with Type 2 Diabetes

**DOI:** 10.3390/jpm12050702

**Published:** 2022-04-28

**Authors:** Fu-Shun Yen, James Cheng-Chung Wei, Ying-Hsiu Shih, Wei-Lin Pan, Chih-Cheng Hsu, Chii-Min Hwu

**Affiliations:** 1Dr. Yen’s Clinic, No. 15, Shanying Road, Gueishan District, Taoyuan 33354, Taiwan; yenfushun@gmail.com; 2Department of Allergy, Immunology & Rheumatology, Chung Shan Medical University Hospital, No. 110, Sec. 1, Jianguo N. Rd., South District, Taichung 40201, Taiwan; jccwei@gmail.com; 3Institute of Medicine, Chung Shan Medical University, No. 110, Sec. 1, Jianguo N. Rd., South District, Taichung 40201, Taiwan; 4Graduate Institute of Integrated Medicine, China Medical University, No. 91, Hsueh-Shih Road, Taichung 40402, Taiwan; 5Management Office for Health Data, China Medical University Hospital, 3F., No.373-2, Jianxing Road, Taichung 40459, Taiwan; hsiu.cmuh@gmail.com; 6College of Medicine, China Medical University, No. 110, Sec. 1, Jianguo N. Rd., South District, Taichung 40201, Taiwan; 7Department of Internal Medicine, Mackay Memorial Hospital, No. 92, Sec. 2, Zhongshan N. Rd., Taipei 10449, Taiwan; kpsmile01058@gmail.com; 8Institute of Population Health Sciences, National Health Research Institutes, 35 Keyan Road, Zhunan, Miaoli County 35053, Taiwan; 9Department of Health Services Administration, China Medical University, No. 91, Hsueh-Shih Road, Taichung 40402, Taiwan; 10Department of Family Medicine, Min-Sheng General Hospital, 168 ChingKuo Road, Taoyuan 33044, Taiwan; 11National Center for Geriatrics and Welfare Research, National Health Research Institutes, 35 Keyan Road, Zhunan 35053, Taiwan; 12Faculty of Medicine, National Yang-Ming Chiao Tung University School of Medicine, No. 155, Sec.2, Linong Street, Taipei 11221, Taiwan; 13Section of Endocrinology and Metabolism, Department of Medicine, Taipei Veterans General Hospital, No. 201, Sec. 2, Shipai Road, Beitou District, Taipei 11217, Taiwan

**Keywords:** type 2 diabetes mellitus, metformin, urinary tract infection, sepsis, death

## Abstract

We conducted this study to compare the morbidity and mortality associated with UTI and sepsis, between metformin users and nonusers in patients with diabetes. As such, 40,774 propensity score-matched metformin users and nonusers were identified from Taiwan’s National Health Insurance Research Database, between 1 January 2000 and 31 December 2017. We adopted the Cox proportional hazards model with robust standard error estimates for comparing the risks of UTI, sepsis, and death due to UTI or sepsis, in patients with T2DM. Compared with the nonuse of metformin, the aHRs (95% CI) for metformin use in UTI, recurrent UTI, sepsis, and death due to UTI or sepsis were 1.06 (0.98, 1.15), 1.08 (0.97, 1.2), 1.01 (0.97, 1.06), and 0.58 (0.42, 0.8), respectively. The cumulative incidence of death due to UTI or sepsis was significantly lower in metformin users than in nonusers (*p* = 0.002). A longer cumulative duration of metformin use had a lower aHR in the risk of death due to UTI or sepsis than metformin nonuse. In patients with T2DM, metformin use showed no significant differences in the risks of UTI, recurrent UTI, or sepsis. However, it was associated with a lower risk of death due to UTI or sepsis than metformin nonuse.

## 1. Introduction

In the 1920s, before the discovery of insulin and antibiotics, the main causes of death among patients with diabetes were ketoacidosis and infectious diseases [1]. Even after the discovery of insulin, infection has remained an important issue, though macrovascular and microvascular diseases are the main complications of diabetes [2]. Patients with diabetes have compromised immune function, disorders of cytokine secretion, impaired bladder emptying due to autonomic neuropathy, and poor wound healing due to vascular insufficiency [3]. Therefore, patients with diabetes may be at a higher risk of urinary tract infection (3.0–4.3-fold), pneumonia (1.2–2.6-fold), cellulitis (1.8–2.0-fold), and sepsis (2.0–3.2-fold) than those without diabetes [2]. They are susceptible to protracted and severe infections, and the risk of infectious hospitalization is also higher than in those without diabetes [2,3]. As patients with diabetes are more prone to vascular complications, they are at higher risk of infection-linked mortality [4].

UTI is the most common infection in patients with diabetes [5,6]. It is an ascending infection, which spreads from the urethra (urethritis) and urinary bladder (cystitis) to the kidney (pyelonephritis). Asymptomatic bacteriuria, defined as ≥10^5^ bacterial colonies per milliliter of urine, is frequently observed (7.4%) in women with diabetes [5,6]. It can progress to symptomatic urinary tract infection (UTI) [5]. Studies show that persons with diabetes are at higher risk of new-onset UTI, recurrence, and hospitalization for UTI than those without diabetes [6,7]. UTI can progress to bacteremia with a poorer prognosis [7,8]. The prevention and mitigation of UTI in patients with diabetes are crucial concerns.

Metformin was introduced as an antimalarial and anti-influenza agent in the 1940s, and it was first launched as a hypoglycemic agent in 1957 [9]. In addition to being a first-line, inexpensive hypoglycemic agent, it has collateral effects of anti-atherosclerosis, anti-tumor, anti-aging, anti-inflammation, and anti-infection [10]. Metformin lowers the risk of pneumonia [11], tuberculosis [12], and sepsis [13], compared with metformin nonuse with diabetes on other treatments. These health-improving effects of metformin use have been contested by previous pharmacoepidemiology studies [14], and few studies have investigated the impact of metformin on UTI. Therefore, we conducted this study to compare the risks associated with UTI, recurrent UTI, sepsis, and mortality due to UTI or sepsis between metformin users and nonusers.

## 2. Materials and Methods

### 2.1. Study Population

The Taiwanese government authorized the Bureau of National Health Insurance in 1995 to design the National Health Insurance (NHI) program as a compulsory insurance system. Approximately 99% of Taiwan’s 23 million people joined the NHI program in 2000 [15]. Information about the insured, including place of residence, age, sex, insurance premium, diagnosis, medications, and medical procedures, is recorded in the NHI Research Database (NHIRD). Diagnosis is based on the International Classification of Diseases, Ninth and Tenth Revision, Clinical Modification (ICD-9-CM and ICD-10-CM) in the inpatient and outpatient claims. The NHIRD is linked to the National Death Registry to provide death information. The study was conducted in accordance with the Declaration of Helsinki, and approved by the Research Ethics Committee of China Medical University and Hospital (CMUH109-109-REC2-031). The identifiable information of caregivers and patients was scrambled and encrypted before release. Informed consent was waived by the Research Ethics Committee.

### 2.2. Study Design

We identified patients with newly diagnosed type 2 diabetes mellitus (T2DM) between 1 January 2000 and 31 December 2017, and followed them up to 31 December 2018. Diagnosis of T2DM was based on ICD codes (ICD-9 code: 250, except 250.1×; ICD-10: E11. Appendix A) for at least 3 outpatient visits or one hospitalization record. The algorithm of using ICD codes to define T2DM was validated by a Taiwan-based study with an accuracy of 74.6% [16]. Patient exclusion criteria (Figure 1) were as follows: (1) missing age or sex data; (2) aged below 20 years or above 80 years; (3) diagnosis of type 1 diabetes, hepatic failure, or history of surgery involving the urinary system; malignant neoplasms of the urinary tract, lymphatic, and hematopoietic tissue; incident dialysis; (4) history of immunosuppressant therapy; (5) the glucagon-like peptide 1 agonists were introduced in 2011, and the sodium-glucose cotransporter 2 inhibitors were launched in Taiwan in 2016. Because the number of patients using these two drugs is very small in the investigated database, we excluded patients using these drugs in this study, or with (6) prior diagnosis of T2DM before 1 January 2000, to exclude prevalent T2DM cases.

### 2.3. Procedures

We defined the first date of metformin use (ATC code: A10BA) as the index date. Patients receiving metformin treatment were study cases, and those who did not receive metformin before the study period served as controls. The index date of the metformin nonusers was assigned as the same date as their corresponding paired metformin users’ index date. Some relevant variates were assessed and matched between metformin users and nonusers, including age, sex, body weight, obesity, severe obesity, smoking status, alcohol disorders, comorbidities of hypertension (HT), dyslipidemia, coronary artery disease (CAD), stroke, atrial fibrillation, heart failure, peripheral arterial occlusive disease (PAOD), chronic kidney disease, retinopathy, chronic obstructive pulmonary disease (COPD), rheumatoid arthritis, systemic lupus erythematosus, hepatitis (including hepatitis B and C infection), liver cirrhosis, urolithiasis, cancers, psychosis, depression, and dementia, diagnosed within 1 year before the index date; medication use, including oral antidiabetic drugs, insulin, statin, corticosteroids, non-steroidal systemic anti-inflammatory drugs (NSAIDs), and aspirin, during the follow-up period. We used the Charlson Comorbidity Index (CCI), Diabetes Complication Severity Index (DCSI) score [17,18], and the number of oral antidiabetic drugs to evaluate T2DM severity.

### 2.4. Main Outcomes

We observed and compared the incidence rates of UTI (by assessing the ICD codings of urethritis, cystitis, acute pyelonephritis, with at least two outpatient visits and antibiotics use, or one hospitalization), recurrent UTI (the second episode of UTI occurring >30 days after the initial episode was considered as recurrence), hospitalization for sepsis (by the discharge diagnosis), and death due to UTI or sepsis (certified by the link to the National Death Registry) between the study and control groups during the follow-up time. The cumulative incidences of UTI, recurrent UTI, sepsis, and death due to UTI or sepsis were compared between metformin users and nonusers.

### 2.5. Statistical Analysis

We used propensity-score matching to optimize the relevant variables between metformin users and nonusers [19]. We estimated the propensity score for each patient using non-parsimonious multivariable logistic regression, with metformin use as the dependent variable; we included 43 clinically related covariates as independent variables (Table 1). The nearest-neighbor algorithm was used to construct matched pairs, assuming the *p*-value > 0.05 to be a negligible difference between the study and comparison cohorts.

Crude and multivariable-adjusted Cox proportional hazards models were used to compare the outcomes between metformin users and nonusers. The results are presented as hazard ratios (HRs) and 95% confidence intervals (CIs) for metformin users compared with nonusers. To calculate the investigated risks, we censored patients on the date of respective outcomes, death, or at the end of the follow-up on 31 December 2018, whichever came first. The Kaplan–Meier method and log-rank tests were used to compare the cumulative incidence of UTI, sepsis, and death due to UTI or sepsis during the follow-up period between metformin users and nonusers. We performed a subgroup analysis for the risk of death due to UTI or sepsis among patients aged 20–60, 61–80 years, females, and males. We also assessed the cumulative duration of metformin use for the risk of death due to UTI or sepsis compared with metformin nonuse.

SAS (version 9.4; SAS Institute, Cary, NC, USA) was used for the statistical analysis, and a two-tailed *p*-value < 0.05 was considered significant.

## 3. Results

### 3.1. Participants

From 1 January 2000 to 31 December 2017, we identified 278,298 patients with newly diagnosed T2DM. Of these, 176,556 were metformin users, and 101,742 were nonusers (Figure 1). After excluding ineligible cases, 1:1 propensity score matching was used to construct 40,774 pairs of metformin users and nonusers. In the matched cohorts (Table 1), 46.73% of patients were female; the mean (SD) age was 56.53 (13.06) years. The mean follow-up time for metformin users and nonusers was 5.19 (3.79) years and 4.60 (4.01) years, respectively.

### 3.2. Main Outcomes

In the matched cohorts (Table 2), 1293 (3.17%) metformin users and 1084 (2.66%) nonusers developed UTI during the follow-up period (incidence rate: 6.14 vs. 5.82 per 1000 person-years). In the multivariable model, metformin users showed no significant differences in the risks of UTI (aHR = 1.06, 95% CI = 0.98–1.15), recurrent UTI (aHR = 1.08, 95% CI = 0.97–1.2), and sepsis (aHR = 1.01, 95% CI = 0.97–1.06) compared to nonusers (Table 2). However, metformin users demonstrated a significantly lower risk of death due to UTI or sepsis (aHR = 0.58, 95% CI = 0.42–0.8) than nonusers (Table 2).

The Kaplan–Meier analysis also showed that the cumulative incidence of death due to UTI or sepsis was significantly lower in metformin users than in nonusers (log-rank test *p* value = 0.002; Figure 2).

### 3.3. Subgroup Analysis

Metformin users aged 20–60 years were associated with significantly lower risk of death due to UTI or sepsis (0.42 (0.18–0.98)); those aged 61–80 years, female, and male were associated with non-significantly lower risk of death due to UTI or sepsis, as compared with metformin nonusers (Table 3).

### 3.4. Cumulative Duration of Metformin Use

We observed the association between the cumulative duration of metformin use and the risk of mortality due to UTI or sepsis (Table 3). A longer cumulative duration (<182, 182–364, >364 days) of metformin use was associated with a lower risk of mortality due to UTI or sepsis (1.17 (0.74–1.85), 1.22 (0.69–2.16), 0.31 (0.2–0.49); (Table 4)).

## 4. Discussion

This study showed that metformin use in patients with T2DM was not associated with significant differences in the risks of UTI, recurrent UTI, and sepsis compared with metformin nonuse. However, metformin use was associated with a significantly lower risk of death due to UTI or sepsis, and a longer duration of metformin use tended to confer a lower risk of mortality due to UTI or sepsis.

UTI is the most common infection in patients with T2DM [3]. It may occur due to compromised cellular and humoral immune function, autonomic neuropathy with impaired bladder emptying, elevated glycosuria to support bacterial growth, and increased Escherichia coli adhesion to uroepithelial cells in patients with T2DM [20]. Patients with T2DM are at higher risk of symptomatic UTI, recurrent UTI, and hospitalization due to UTI than those without T2DM [2,3,6,7,8]. Acute pyelonephritis, bacteremia, and complicated urinary tract infections, such as emphysematous pyelonephritis, nephric and perinephric abscess, are common in patients with T2DM (probably due to microvascular complications associated with reduced renal blood flow) [3,5,6,8]. Studies have shown that metformin may attenuate tuberculosis [12] and pneumonia risks [11]. However, there is no known study to compare the risk of UTI between metformin users and nonusers. Our study showed no significant difference in UTI and recurrent UTI risk between metformin use versus nonuse. As laboratory data from patients were unavailable, we could not investigate the impact of metformin use on the risk of asymptomatic bacteriuria. We defined UTI as at least two outpatient visits and antibiotics use or one hospitalization. Patients with UTI who just received one outpatient claim, or UTI treated by self-care would not be censored in this study. However, this potential underestimation of UTI could non-differentially occur in the study and control groups, resulting in a bias of associations toward the null [7].

UTI may progress to pyelonephritis, bacteremia, and even sepsis [6,8]. Sepsis is a life-threatening multi-organ dysfunction, caused by a dysregulated host response to infections. Patients with T2DM are at higher risk of hospitalization due to sepsis (aHR 2.21 (2.07–2.36)) than those without T2DM [21]. A nested case-control study showed that metformin reduced the risk of sepsis [13]. However, our study showed no significant differences in the risk of sepsis between metformin users and nonusers. The conflicting results between these two studies may be due to different study designs. The event rate of sepsis in this study is higher than that of urinary tract infection, possibly because sepsis can occur due to various causes, such as urinary tract infection, pneumonia, and cellulitis.

Reports show that metformin can reduce the risk of mortality due to COVID-19 infection [22], pneumonia [12], and sepsis [23]. This study showed that metformin use was associated with a 42% lower risk of mortality due to UTI or sepsis, and a longer duration of metformin use had a lower mortality risk. Furthermore, the subgroup analysis revealed that metformin users aged 20–40 years had a significantly lower risk of death due to UTI or sepsis than nonusers. This may indicate that the beneficial effect of metformin on death risk due to UTI or sepsis is more prominent in younger patients with T2DM.

The possible mechanisms for metformin to attenuate the risk of mortality due to UTI or sepsis may be: (1) Metformin can inhibit mitochondrial respiratory-chain complex-1 and activate the adenosine monophosphate (AMP)-activated protein kinase (AMPK) pathway to facilitate neutrophil activation, chemotaxis, and bacterial killing [22,24]. (2) Metformin can improve T and B-cell function in patients with obesity and T2DM [24,25]. (3) Metformin can decrease pro-inflammatory markers of C-reactive protein, interferon-α [26], tumor necrosis factor-α, and interleukin-6, and increase the level of the anti-inflammatory marker IL-10 [22,24]. (4) The inhibition of mitochondrial complex-1 and electron transport by metformin can decrease the energy supply required for bacterial growth. Metformin can also inhibit bacterial gluconeogenesis, and limited glycerol use in the Krebs cycle can decrease bacterial virulence. The anti-folate effect of metformin may inhibit the bacterial folate cycle and suppress bacterial growth [27]. (5) Metformin decreases the expression of nitric oxide synthase and ameliorates vasodilatation with anti-endotoxaemic and vasoactive properties [23]. We need to perform further studies to realize the beneficial role of metformin on the risk of death due to UTI or sepsis. That is, to know whether the protective effect of metformin is due to improved glucose, insulin resistance, cellular or humoral immunity in patients with T2DM.

The strength of this study is that it is a nationwide, population-based study, with the observational time spanning 17 years and the availability of substantial information on demographics, comorbidities, and medications that may contribute to outcomes. This study also has some limitations. First, this dataset lacks information on diet and exercise, smoking and alcohol drinking habits, and family history. The NHIRD does not provide data on urine tests, urine and blood cultures, renal function and biochemical tests, hemoglobin A1C, and immune functional tests, which may preclude an accurate evaluation of immune function, UTI, bacteremia, sepsis, and T2DM severity. However, we matched several critical variables, such as age, sex, comorbidities, CCI, and medications for maximal balance between the condition between study and control groups. We also matched the item and number of oral antidiabetic drugs, insulin use, and DCSI scores to balance the severity of T2DM and increase their comparability. Second, this study mainly observed a Taiwanese population, and the result may not apply to other racial and ethnic groups. Third, the success of UTI treatment is related to the resistance pattern of antibiotics, but this dataset does not have data on the antibiotic resistance. Fourth, patients with advanced kidney, liver, or heart diseases may discontinue metformin use to avoid lactic acidosis and mortality. We excluded patients with hemodialysis or hepatic failure and matched the comorbidities of chronic kidney disease, heart failure, alcohol-related disorders, and liver cirrhosis for further analysis to avoid the bias of confounding by indication. Finally, a cohort study is usually associated with bias due to some uncovered and unobserved confounding factors, and a randomized controlled trial is warranted to verify our study.

## 5. Conclusions

Though macrovascular and microvascular complications are the main complications of diabetes, infections and sepsis remain critical concerns, with limited relevant guidelines and advice. UTI is the most frequent complication, and sepsis is a life-threatening infection in patients with T2DM. Our study demonstrated that metformin use showed no significant difference in the risks of UTI and sepsis. However, it showed a significantly lower risk of death due to UTI or sepsis. Metformin may play a role in reducing mortality due to UTI or sepsis. The anti-phlogistic and antimicrobial effects of metformin need further investigation to repurpose the drug into a broader spectrum of use.

## Figures and Tables

**Figure 1 jpm-12-00702-f001:**
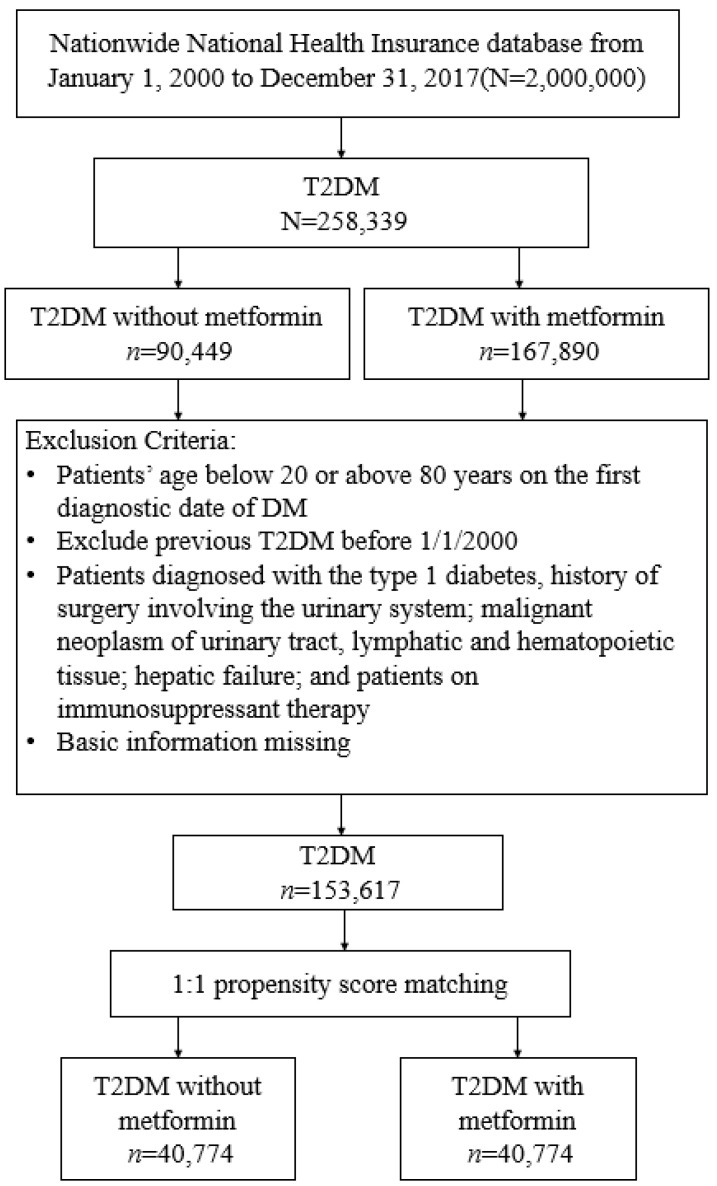
Flow chart of the identified process in this study.

**Figure 2 jpm-12-00702-f002:**
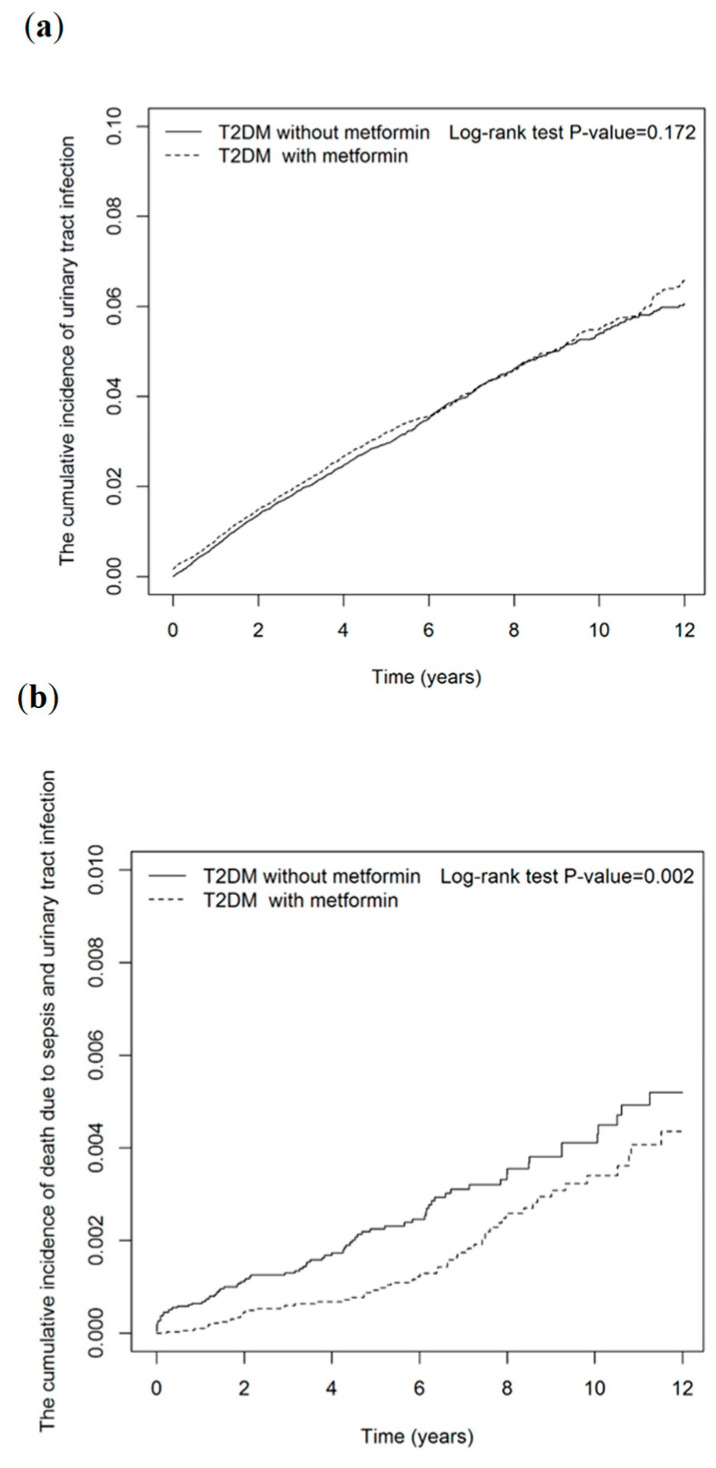
Cumulative incidences of (**a**) urinary tract infection (UTI), (**b**) death due to UTI or sepsis, for metformin users vs. metformin nonusers.

**Table 1 jpm-12-00702-t001:** Comparison of baseline characteristics in T2DM cohorts with and without metformin use.

Variables	T2DM without Metformin		T2DM with Metformin		*p*
	(*N* = 40,774)		(*N* = 40,774)		
	*n*	%	*n*	%	
Sex					0.123
female	18,942	46.46	19,162	47.00	
male	21,832	53.54	21,612	53.00	
Age					0.164
20–40	5049	12.38	5228	12.82	
41–60	19,158	46.99	19,093	46.83	
61–80	16,567	40.63	16,453	40.35	
mean, (SD) †	56.62	13.03	56.43	13.09	0.040
Comorbidities					
Hypertension	22,445	55.05	23,020	56.46	<0.001
Dyslipidemia	23,577	57.82	23,990	58.84	0.003
Coronary artery disease	10,963	26.89	10,964	26.89	0.994
Stroke	3645	8.94	3719	9.12	0.366
Atrial fibrillation	41	0.10	44	0.11	0.745
Heart failure	2715	6.66	2765	6.78	0.484
PAOD	1401	3.44	1363	3.34	0.462
CKD	2028	4.97	1867	4.58	0.008
Retinopathy	2752	6.75	2687	6.59	0.362
COPD	10,570	25.92	10,728	26.31	0.208
Rheumatoid arthritis	729	1.79	699	1.71	0.423
Systemic lupus erythematosus	69	0.17	67	0.16	0.864
Hepatitis B or C infection	3493	8.57	3411	8.37	0.302
Liver cirrhosis	859	2.11	868	2.13	0.827
Urolithiasis	3890	9.54	3856	9.46	0.685
Cancers	1389	3.41	1377	3.38	0.816
Psychosis	832	2.04	880	2.16	0.241
Depression	13,147	32.24	13,148	32.25	0.994
Dementia	1136	2.79	1120	2.75	0.733
Obesity	883	2.17	898	2.20	0.719
Smoking	1073	2.63	1093	2.68	0.663
Alcoholic diseases	2309	5.66	2397	5.88	0.186
CCI					<0.001
1	9406	23.07	8924	21.89	
2–3	20,808	51.03	21,396	52.47	
>3	10,560	25.90	10,454	25.64	
DCSI					0.082
0	16,557	40.61	16,286	39.94	
1	7649	18.76	7851	19.25	
≥2	16,568	40.63	16,637	40.80	
Medication					
SU	4342	10.65	4487	11.00	0.102
TZD	364	0.89	369	0.91	0.853
DPP-4i	406	1.00	402	0.99	0.888
AGI	926	2.27	949	2.33	0.591
OAD drugs					0.503
1	40,132	98.43	40,096	98.34	
2–3	633	1.55	671	1.65	
>3	9	0.02	7	0.02	
Insulin	13,624	33.41	13,838	33.94	0.113
Corticosteroid	422	1.04	411	1.01	0.702
Statin	12,358	30.31	12,364	30.32	0.964
NSAIDs	39,998	98.10	40,315	98.87	<0.001
Aspirin	14,514	35.60	14,611	35.83	0.478

T2DM, type 2 diabetes mellitus; SD, standard deviation; PAOD, peripheral arterial occlusive disease; CKD, chronic kidney disease; COPD, chronic obstructive pulmonary disease; SU, sulfonylureas; TZD, thiazolidinedione; DPP-4i, dipeptidyl peptidase-4 inhibitor; AGI, alpha-glucosidase inhibitors; NSAIDs, non-steroidal anti-inflammatory drugs. Data shown as *n* (%) or mean ± SD. †: Student’s *t*-test.

**Table 2 jpm-12-00702-t002:** Hazard ratios (HRs) and 95% confidence intervals (CIs) for outcomes among the sampled patients.

Outcome	T2DM without Metformin			T2DM with Metformin								
	*n*	PY	IR	*n*	PY	IR	cHR	(95% CI)	*p*-Value	aHR†	(95% CI)	*p*-Value
UTI	1084	186,381	5.82	1293	210,672	6.14	1.06	(0.98, 1.15)	0.1728	1.06	(0.98, 1.15)	0.1732
Recurrence of UTI	642	189,353	3.39	786	214,173	3.67	1.08	(0.97, 1.2)	0.1425	1.08	(0.97, 1.2)	0.1676
Hospitalization for sepsis	3579	182,854	19.57	4141	203,933	20.31	1.05	(1, 1.1) *	0.0431	1.01	(0.97, 1.06)	0.5882
Death (UTI _Sepsis)	93	192,448	0.48	62	218,118	0.28	0.6	(0.44, 0.83) **	0.0019	0.58	(0.42, 0.8) ***	<0.001

PY: person-years; IR: incidence rate, per 1000 person-years; cHR, crude hazard ratio; aHR: adjusted hazard ratio. aHR†: multivariable analysis including sex, age, comorbidities, CCI, DSCI, corticosteroid, statin, NSAIDs, aspirin, insulin, item and number of oral antidiabetic drugs. * *p*-value < 0.05; ** *p* < 0.01, *** *p* < 0.001.

**Table 3 jpm-12-00702-t003:** Risk of death due to UTI or sepsis in metformin users vs. nonusers in patients with T2DM stratified by age and sex.

	Death (UTI_Sepsis)											
	Non-Metformin			Metformin			Univariate			Multivariate		
Variables	*n*	PY	IR	*n*	PY	IR	cHR	(95% CI)	*p*-Value	aHR†	(95% CI)	*p*-Value
Age												
20–60	17	110,788	0.15	8	120,007	0.07	0.44	(0.19, 1.02)	0.056	0.42	(0.18, 0.98) *	0.045
61–80	65	65,371	0.99	63	77,931	0.81	0.81	(0.58, 1.15)	0.245	0.82	(0.58, 1.17)	0.277
Sex												
female	35	87,446	0.40	32	97,257	0.33	0.83	(0.52, 1.35)	0.454	0.69	(0.42, 1.12)	0.132
male	47	88,713	0.53	39	100,682	0.39	0.75	(0.49, 1.15)	0.186	0.8	(0.52, 1.23)	0.310

PY: person-years; IR: incidence rate, per 1000 person-years; cHR, crude hazard ratio; aHR: adjusted hazard ratio. aHR†: multivariable analysis including sex, age, comorbidities, and medications in Table 1. *: *p*-value < 0.05.

**Table 4 jpm-12-00702-t004:** Hazard ratio of death due to UTI or sepsis for stratification by the cumulative use of metformin.

Variables	Death (UTI or Sepsis)						
	N	PY	IR	cHR	(95% CI)	aHR†	(95% CI)
Non-use of metformin	93	192,448	0.48	1.00	(Reference)	1.00	(Reference)
Cumulative metformin use (days)							
<182	24	38,138	0.63	1.41	(0.89, 2.24)	1.17	(0.74, 1.85)
182–364	14	24,515	0.57	1.25	(0.68, 2.29)	1.22	(0.69, 2.16)
>364	24	155,465	0.15	0.5	(0.34, 0.74) ***	0.31	(0.2, 0.49) ***

PY: person-years; IR: incidence rate, per 1000 person-years; cHR, crude hazard ratio; aHR: adjusted hazard ratio. aHR†: multivariable analysis including sex, age, comorbidities, CCI, DCSI, corticosteroid, statin, NSAIDs, aspirin, insulin, item and number of oral antidiabetic drugs. *** *p* < 0.001.

## Data Availability

Data from this study are available from the National Health Insurance Research Database (NHIRD), published by the Taiwan National Health Insurance (NHI) Administration. The data utilized in this study cannot be made available in the paper, the Appendix A, or in a public repository, due to the “Personal Information Protection Act” executed by the Taiwanese government, starting from 2012. Requests for data can be sent as a formal proposal to the NHIRD Office (https://dep.mohw.gov.tw/DOS/cp-2516-3591-113.html accessed on 25 March 2022) or by email to stsung@mohw.gov.tw.

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
