# Peer review of "Role of Metformin in Morbidity and Mortality Associated with Urinary Tract Infections in Patients with Type 2 Diabetes"

_jpm, 2022, doi:10.3390/jpm12050702_

Round 1
Reviewer 1 Report
The Yen et al., 2022, Manuscript ID: jpm-1679349 addresses role of diabetic drug metformin in morbidity and mortality associated with urinary tract infections in patients with T2D. A search on Pubmed.gov for the terms "urinary tract infections" and "Metformin" and "type 2 Diabetes" keywords resulted in only 44 hits that depicts the novelty of this study.
The authors showed that metformin use showed no significant difference in the risks of UTI and sepsis but still there are few queries and few suggestion which makes this manuscript more representable to be publish.
- Why the authors have not shown the changes in glucose level in the table 1 where they showed Comparison of baseline characteristics in T2DM cohorts with and without metformin use?
- Can the authors justify the effect of metformin in reducing the risks of UTI and sepsis is dependent on the age factor? Do they found same pattern in the age of patients?
- Did the authors found any changes in insulin resistance index in with and without metformin treated T2D patients?
- Do the authors have any plan to conduct the research to know the molecular mechanisms of beneficial role of metformin in the UTI disease?
Author Response
Dear Academic Editor:
Please find attached a revised version of our document “Role of Metformin in Morbidity and Mortality Associated with Urinary Tract Infections in Patients with Type 2 Diabetes”. We would like to resubmit for publication as an original article in Journal of Personalized Medicine.
Your comments and those of the reviewers were highly insightful and enabled us to improve the quality of our document. In the following pages are our responses to each comment from the reviewer(s) as well as your own comments.
Revisions in the text are shown [yellow highlights]. We hope that our revisions to the document combined with our accompanying responses will be sufficient to render our document suitable for publication in Journal of Personalized Medicine.
We look forward to hearing from you soon.
Yours sincerely,
Chih-Cheng Hsu
Institute of Population Health Sciences, National Health Research Institute
Tel.: +886-37-246166 #36336
Fax: +886-37-586261
Email: cch@nhri.edu.tw
Address: 35 Keyan Road, Zhunan, Miaoli County 35053, Taiwan.
Chii-Min Hwu
Faculty of Medicine, National Yang-Ming Chiao Tung University School of Medicine
Tel.: +886-2-28757516
Fax: +886-2-28751429
E-Mail: chhwu@vghtpe.gov.tw
Address: No.155, Sec.2, Linong Street, Taipei 11221, Taiwan
Responses to the comments of Reviewer #1
The Yen et al., 2022, Manuscript ID: jpm-1679349 addresses role of diabetic drug metformin in morbidity and mortality associated with urinary tract infections in patients with T2D. A search on Pubmed.gov for the terms "urinary tract infections" and "Metformin" and "type 2 Diabetes" keywords resulted in only 44 hits that depicts the novelty of this study.
The authors showed that metformin use showed no significant difference in the risks of UTI and sepsis but still there are few queries and few suggestion which makes this manuscript more representable to be publish.
- Why the authors have not shown the changes in glucose level in the table 1 where they showed Comparison of baseline characteristics in T2DM cohorts with and without metformin use?
Response: Thank you for your encouragement and reviewing of our manuscript. Because the National Health Insurance Research Database (NHIRD) lacks data of glucose and hemoglobin A1C levels, therefore we cannot show the change of glucose levels in Table 1. We have mentioned this limitation on page 9, and matched the item and number of oral antidiabetic drugs, insulin use, and DCSI scores trying to balance the severity of T2DM between metformin users and nonusers.
- Can the authors justify the effect of metformin in reducing the risks of UTI and sepsis is dependent on the age factor? Do they found same pattern in the age of patients?
Response: We have performed a subgroup analysis of patients with different age on the risk of death due to UTI or sepsis. At first, we divided the age groups into three groups for analysis (attached table). However, the event rate in the 20-40 age group was too small to analyse. Therefore, we divided the patients into two groups, age 20-60 and 61-80 years. Both groups of metformin users had a lower risk of death due to UTI or sepsis than metformin nonusers. But, in the 20-40 age group of metformin users, the risk of death was significantly lower than metformin nonusers. This may indicate that the beneficial effect of metformin in the risk of death due to UTI or sepsis is more prominent in younger patients with T2DM. We have added the statement of subgroup analysis in the section of Methods, Results and Discussion.
- Did the authors found any changes in insulin resistance index in with and without metformin treated T2D patients?
Response: Because the administrative dataset does not have data of glucose and insulin levels, so we cannot explore the change in insulin resistance index between metformin users and nonusers.
- Do the authors have any plan to conduct the research to know the molecular mechanisms of beneficial role of metformin in the UTI disease?
Response: Thank you for your recommendation. We have added the statement of future plans to know the molecular mechanisms of beneficial role of metformin in the risk of death due to UTI or sepsis on page 8 as” We need to perform further studies to realize the beneficial role of metformin on the risk of death due to UTI or sepsis. That is to know the protective effect of metformin is due to improved glucose, insulin resistance, cellular or humoral immunity in patients with T2DM “.

Reviewer 2 Report
In this study the role of metformin was evaluated to estimate its effect associated with urinary tract infections in patients with T2D.
The results of this study indicate that the use of metformin in patients with T2D showed no significant difference in the risk of UTI or sepsis. However, it showed a much lower risk of death due to UTI or sepsis and therefore metformin may play a very important role in reducing mortality due to UTI or sepsis.
Calculation of HRs and cumulative incidence is requested, separating males and females.
Author Response
Dear Academic Editor:
Please find attached a revised version of our document “Role of Metformin in Morbidity and Mortality Associated with Urinary Tract Infections in Patients with Type 2 Diabetes”. We would like to resubmit for publication as an original article in Journal of Personalized Medicine.
Your comments and those of the reviewers were highly insightful and enabled us to improve the quality of our document. In the following pages are our responses to each comment from the reviewer(s) as well as your own comments.
Revisions in the text are shown [yellow highlights]. We hope that our revisions to the document combined with our accompanying responses will be sufficient to render our document suitable for publication in Journal of Personalized Medicine.
We look forward to hearing from you soon.
Yours sincerely,
Chih-Cheng Hsu
Institute of Population Health Sciences, National Health Research Institute
Tel.: +886-37-246166 #36336
Fax: +886-37-586261
Email: cch@nhri.edu.tw
Address: 35 Keyan Road, Zhunan, Miaoli County 35053, Taiwan.
Chii-Min Hwu
Faculty of Medicine, National Yang-Ming Chiao Tung University School of Medicine
Tel.: +886-2-28757516
Fax: +886-2-28751429
E-Mail: chhwu@vghtpe.gov.tw
Address: No.155, Sec.2, Linong Street, Taipei 11221, Taiwan
Responses to the comments of Reviewer #2
In this study the role of metformin was evaluated to estimate its effect associated with urinary tract infections in patients with T2D.
The results of this study indicate that the use of metformin in patients with T2D showed no significant difference in the risk of UTI or sepsis. However, it showed a much lower risk of death due to UTI or sepsis and therefore metformin may play a very important role in reducing mortality due to UTI or sepsis.
- Calculation of HRs and cumulative incidence is requested, separating males and females.
Response: Thank you for your encouragement and suggestions. We have performed the subgroup analysis, calculated HRs and cumulative incidences of death due to UTI or sepsis of male and female in metformin users and nonusers The female and male metformin users had non-significantly lower HRs and cumulative incidence of death due to UTI or sepsis than metformin nonusers (attached Table and Figures). We have added the results of subgroup analysis in the section of Methods (page 4) and Results (page 7-8).
